# Healthcare resource use and associated costs in a cohort of hospitalized COVID-19 patients in Spain: A retrospective analysis from the first to the third pandemic wave. EPICOV study

Georgina Drago[1]*, Francisco Javier Pérez-Sádaba[2], Susana Aceituno[2], Carla Gari[2], Juan Luis López-Belmonte[3]

1 Market Access, Sanofi, Barcelona, Spain, 2 Outcomes' 10 SLU, Castellón de la Plana, Castellón, Spain, 3 Market Access, Sanofi, Madrid, Spain

* georgina.drago@sanofi.com

## Abstract

### Objectives

Describe healthcare resource use and costs per hospitalized coronavirus disease-2019 (COVID-19) patient during the three main outbreak waves.

### Methods

A retrospective observational study. COVID-19 patient data were collected from a dataset from 17 hospitals in the HM Hospitals Group. Mean total costs per hospitalized patient and per day were estimated in each wave, as defined by the Spanish National Health System perspective. In addition, costs were estimated for both patients admitted and those not admitted to the intensive care unit (ICU) and were stratified by age groups.

### Results

A total of 3756 COVID-19 patients were included: 2279 (60.7%) for the first, 740 (19.7%) for the second, and 737 (19.6%) for the and third wave. Most (around 90%) did not require ICU treatment. For those patients, mean ± SD cost per patient ranged from €10 196.1 ± €7237.2 (mean length of stay [LOS] ± SD: 9.7 ± 6.2 days) for the second wave to €9364.5 ± €6321.1 for the third wave (mean 9.0 ± 5.7 days). Mean costs were around €1000 per day for all the waves. For patients admitted to the ICU, cost per patient ranged from €81 332.5 ± €63 725.8 (mean 31.0 ± 26.3 days) for the second wave to €36 952.1 ± €24 809.2 (mean 15.7 ± 8.2 days) for the third wave. Mean costs per day were around €3000 for all the waves. When estimated by age, mean LOS and costs were greater in patients over 80 when not admitted to the ICU and for patients aged 60 to 79 when admitted to the ICU.

### Conclusions

LOS was longer for patients admitted to the ICU (especially in the first two waves) and for older patients in our study cohort; these populations incurred the highest hospitalization costs.

**Data Availability Statement:** All relevant data are within the paper and its Supporting Information files.

**Funding:** This study was funded by Sanofi. G.D. and J.L.L-B are Sanofi employees, and they played a role in the study design and manuscript preparation. Outcomes'10 was funded by Sanofi to support the study design, analysis, and manuscript preparation.

**Competing interests:** G.D. and J.L.L-B. declare that are employees of Sanofi and may hold shares and/or stock options in the company. C.G., F.J.P-S. and S.A. declare that they work for an independent research entity and that they have received fees for their contribution to project development and the writing of the manuscript.

## Introduction

The coronavirus disease-2019 (COVID-19) pandemic caused by the respiratory syndrome coronavirus 2 (SARS-CoV-2) infection has represented a great clinical burden. Approximately 10% of infected patients developed severe acute respiratory syndrome (SARS) and required hospitalization, with mortality rates reaching between 2% and 5% [1, 2]. Spain was one of the most seriously affected countries, especially during the first outbreak wave of the pandemic, starting in February 2020 and lasting until June 2020 (five months) and leading to a nation-wide lockdown [3]. In this outbreak wave, the seroprevalence of SARS-CoV-2 was as high as 5.0% (95% CI 4.7–5.4) of the population and exceeded 10% in certain regions, such as Madrid [4]. In this scenario, it is hardly surprising that the pandemic exerted great pressure on the Spanish national healthcare system and on society [5, 6].

As the incidence of SARS-CoV-2 infection increased exponentially in March 2020 in Spain, the number of infected patients requiring hospitalization began to exceed hospital capacity in many public hospitals. This critical situation forced the Spanish authorities to take extraordinary measures to decongest public hospitals, such as relocating patients from the public to private healthcare facilities or creating temporary healthcare structures to absorb the surplus of patients [7, 8]. Subsequently, two more outbreak waves followed at the national level: the second occurred from summer (July 2020) until early December 2020 (five months) [9] and the third started after the Christmas season in 2020 and lasted until the vaccination campaigns in February 2021 (two months) [10]. From then on, there was a rebound of infections in the spring of 2021, but the pressure on hospitals was much lower than during the previous waves [11].

Over this period of approximately one year (from March 2020 to February 2021), healthcare providers were forced to confront an unknown disease that often required hospitalization and complex management. These included ventilation support and the use of multiple treatments, usually without sufficient evidence of their effectiveness or safety [12]. However, the hospital management of COVID-19 patients evolved rapidly over the first months with continuous learning from clinical experience and the constant emergence of new clinical evidence [13].

In order to understand the impact on healthcare systems during the first year of the pandemic, it is essential to know how hospital expenses were allocated and how they changed during the different outbreak waves. Average healthcare hospital costs of COVID-19 are available in different settings [14–19]; however, so far, none of these cost analyses have taken into account the evolving nature of the COVID-19 pandemic from its onset until the systematic vaccination campaigns in 2021. In the present study, we aimed to use real-world data to describe the use of healthcare resources and the associated costs in a cohort of hospitalized patients due to SARS-CoV-2 during three different outbreak waves from the perspective of Spain's national health system (NHS).

## Material and methods

### Design

A retrospective, descriptive, observational study was carried out based on a dataset of patients with SARS-CoV-2 in Spain: "COVID Data Saves Lives" [20]. This dataset includes extensive anonymized information from hospitalized COVID-19 patients admitted to 17 tertiary hospitals belonging to the HM Hospitals Group (Spanish consortium of private hospitals) and amounting to over 1400 beds, located across four Autonomous regions in Spain: Autonomous Community of Madrid, Castilla and Leon, Catalonia, and Galicia [21]. Due to the pandemic situation, these hospitals stopped their scheduled private care activity to attend to emergency cases referred from public centers that could no longer admit patients. Patients were included

for the analysis if they were admitted with confirmed SARS-CoV-2 infection by real-time reverse transcription-polymerase chain reaction (PCR) and followed-up between 31st January 2020 and 13th February 2021 in any of the HM Hospitals participating in the study. In addition, the reasons for patient admission and discharge must be available. The study period was divided into the three main waves of SARS-CoV-2 incidence in Spain, also known as outbreak waves. The data were grouped according to the RENAVE epidemiological surveillance system into three periods corresponding to the different waves as follows: first wave, from 31st January 2020 to 21st June 2020 [22, 23]; second wave, from 22nd June 2020 to 6th December 2020 [9, 23]; and third wave, from 7th December 2020 to 13th February 2021 [10, 23], as far as data in our database were available.

During the study period (31st January 2020 to 13th February 2021), anonymized data from 4475 COVID-19 patients were recorded in the "COVID Data Saves Lives" dataset [20]. All patients were admitted through the emergency department (ED). Dataset was extracted from the Electronic Health Record (EHR) system of the HM Hospitals. The information was organized in tables or datasets including information about the COVID-19 treatment process, complete information on admission and diagnoses treatments, ICU admissions, diagnostic imaging tests, laboratory results, drug administration, and cause of discharge/death. All datasets are linked by a unique admission identifier as a de-anonymization key, created for this purpose, and dissociated from each admission identifier. Among this population, 3756 patients met the study inclusion criteria: n = 2279 were admitted during the first wave, n = 740 during the second, and n = 737 during the third outbreak wave of SARS-CoV-2 (**Fig 1**).

The study protocol was approved by the Ethics Committees of the HM Hospitals Group. The collection of patients' written informed consent was not required as patient data were anonymized and gathered retrospectively.

The present study is reported according to the Reporting of Observational Studies in Epidemiology (STROBE) checklist (**S1 Table**) [24].

## Variables

The following patients' demographic and clinical variables were extracted from the HM Hospitals Group's "COVID Data Saves Lives" dataset [20]: age, gender, vital signs recorded at the

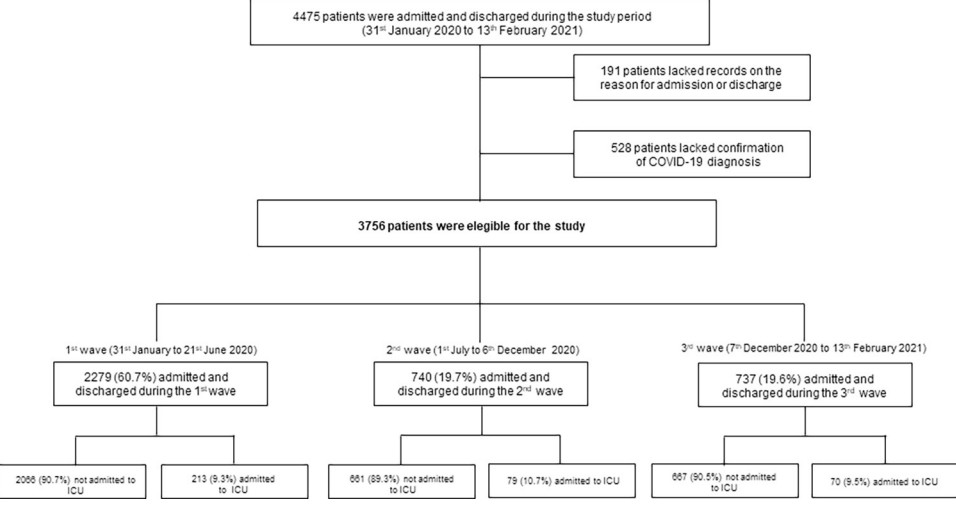

**Fig 1. Flow chart of the study cohort.** Abbreviations: COVID-19 (coronavirus disease-2019); ICU (intensive care unit).

ED (temperature, heart rate, blood pressure, oxygen saturation, and requirement for mechanical ventilation), primary diagnosis at the ED and at hospital admission, secondary diagnosis at hospital admission, and deaths during stay in hospital wards and intensive care units (ICU).

Diagnoses were identified by the 10[th] revision of the International Statistical Classification of Diseases and Related Health Problems codes (ICD-10) [25]. Primary diagnoses at admission corresponded with the COVID-19-diagnosis codes. Because the COVID-19-related ICD-10 (U07.1) took effect as of the second wave, non-specific codes were assigned to COVID-19 (i.e., other viral pneumonia [J12.89], or unspecified pneumonia [J18.9]) during the first wave. In contrast, secondary diagnoses at admission corresponded with either the symptoms or consequences of COVID-19 or the patient's comorbidities. The following healthcare resources were retrieved from the HM Hospitals dataset: procedures at ED, hospital length of stay (LOS), laboratory tests during hospitalization (both in hospital wards and ICU), procedures during hospitalization (both in hospital wards and ICU) and pharmacological treatments during hospitalization in hospital wards (not available for ICU stays). Procedures were identified by the ICD-10 diagnosis and procedure codes [25] whereas the treatments were classified by the Anatomical Therapeutic Chemical-4 (ATC4) Classification system [26]. Laboratory tests were identified in a Spanish healthcare cost lists database [27].

## Analysis

Descriptive statistics were used to summarize patients' sociodemographic and clinical characteristics at the ED and hospital admission, in addition to healthcare resources used and in-hospital mortality. Absolute and relative frequencies were estimated for qualitative variables (gender [male; female], primary and secondary diagnoses, temperature [$> 38˚C$; $\leq 38˚C$], oxygen saturation [$< 95\%$, $\geq 95\%$], the requirement for mechanical ventilation [yes; no], laboratory tests, ED and hospital procedures, pharmacological treatments, or in-hospital mortality). At the same time, measures of central tendency and dispersion (mean, standard deviation [SD]) were used for quantitative variables (age, heart rate, blood pressure, hospital, and ICU length of stay, added to laboratory tests and procedures required per patient). To estimate the mean number of procedures and laboratory tests per hospitalized COVID-19 patient (either per patient who was admitted to the ICU, or per patient who was not admitted to the ICU), we added up the number of the resources registered for each patient during the study period. The mean number of procedures or laboratory tests per patient was then obtained by dividing individual patient numbers by the number of patients in each wave. To estimate the number of laboratory tests, we considered them on a disaggregated level (e.g., "red blood cell counts" rather than "blood count").

The primary outcome of the analysis was the use of healthcare resources and its associated costs per hospitalized COVID-19 patient in the different outbreak waves. As the pharmacological treatment was not available during the ICU stays, we conducted the cost analysis in two separate groups of patients: patients who received care exclusively in the hospital ward and, therefore, had completed treatment data records; and patients who were admitted to the ICU at some point during the hospital stay.

To estimate the mean costs per hospitalized COVID-19 patient (either per patient who was admitted or per patient who was not admitted to the ICU), we calculated total healthcare costs for each patient by estimating the sum of the costs of each resource considered during hospitalization. This estimation was done by multiplying the frequency of use of each resource by its unitary costs in the different outbreak waves. Mean costs per patient were then obtained by the sum of individual patient costs divided by the number of patients in each wave. The costs per patient and day were also estimated. First, individual costs per hospitalized COVID-19 patient

were divided by the days spent in hospital. Average cost per patient and day was then obtained from all the individual costs per day. For the main cost results, in order to show how the costs are distributed in our population, we also estimate the median cost.

The cost analysis was conducted from the perspective of the Spanish National Health System (NHS); thus, unit costs for procedures, laboratory tests, hospital, ICU stay, and treatments were derived from official local sources, reported in euros (€) and update to 2021. Specifically, unitary costs for procedures were based on the Diagnosis Related Groups (DRGs) costs for Spanish General Hospitals or from Autonomous Region's tariffs when DGRs were not available [27], whereas pharmacological costs corresponded to the official selling price or PVL (*Precio de Venta Libre*) published in the BotPlus database [28]. Laboratory tests and hospital stay tariffs varied among the Autonomous Regions [27]; therefore, mean costs among different regions were estimated for each resource to obtain representative costs at a national level. In addition, to estimate laboratory test costs, we considered costs for aggregate tests (e.g., the cost of a blood count instead of the cost of the different components separately) since the total cost of aggregated tests was not deemed to significantly vary, regardless of the number of individual assessments they included.

Both mean costs per patient and cost per patient and day were stratified by the vaccination age bands in Spain: < 12, 12–19, 20–29, 30–39, 40–49, 50–59, 60–69, 70–79, and > 80 years [29]. Additionally, costs per patient were also stratified by the different cost types: hospital stay, treatment, laboratory tests, and procedures.

To identify cost drivers of the covariates in the study population a Generalized Linear Model (GLM) with gamma distribution and log link with stepwise algorithm was used. The stepwise regression consists of iteratively adding and removing predictors, in the predictive model, in order to find the subset of variables in the data set resulting in the best performing model, that is a model that lowers prediction error. Outliers in the residuals of the selected model were removed using the interquartile range. In this process, total cost was selected as the dependent variable, and gender, age, wave, and length of stay were independent variables. The estimated coefficients were converted into cost ratios that can be interpreted as a ratio of adjusted costs between the category of interest versus the reference category for binary predictors or as the percentage increase in average cost per unit increase in a continuous covariate. A p-value of less than 0.05 was considered statistically significant. Two modelling runs were performed, one for ICU patients and one for non-ICU patients.

A deterministic sensitivity analysis was conducted to estimate the impact of selecting particular costs rather than others in the cost analysis results. For this analysis, only laboratory tests and hospital stay were considered as the costs as these resources varied among the different Autonomous Regions. For the base case, we applied the mean costs among the different tariffs available. Meanwhile, for the sensitivity analysis, we evaluated two different scenarios: one including the minimum individual costs for laboratory tests and hospital stay resources and the other with the respective maximum costs reported among the Autonomous Regions.

The data analysis was performed using the STATA version 14 statistical software package.

## Results

### Cohort description

A total of 3756 patients with confirmed COVID-19 diagnosis were admitted and discharged during the study period (**Fig 2**), of which 60.7% were admitted during the first wave, 19.7% during the second and 19.6% during the third outbreak wave. Of the hospitalized patients included in the study, around 10% required ICU admission at some point of the hospital stay (**Table 1**).

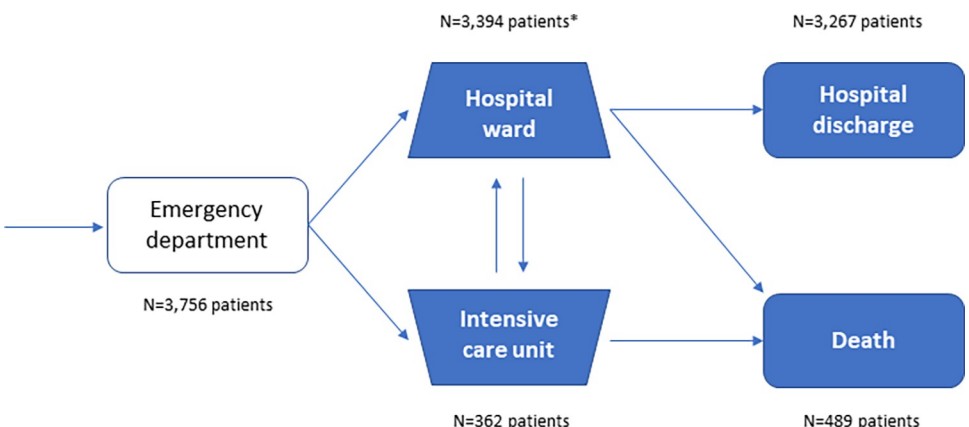

**Fig 2. Inpatient pathway of the study cohort.** *Patients who were only admitted to the hospital ward.

The mean ± SD age of COVID-19 hospitalized patients was 67.9 ± 16.2 years in the first wave, 65.6 ± 16.8 years in the second, and 65.4 ± 16.6 in the third wave, with the population ≥60 accounting for more than two-thirds of the total, and that over 80 ranging from 20.9% in the third wave to 26.3% in the first wave (**S1 Fig**). Regarding the gender, the proportion of men was approximately 60% in all three waves. At the ED, most patients presented fever (temperature > 38˚C), and around half had an oxygen saturation < 95% across the different outbreak waves. The requirement for mechanical ventilation was particularly higher in those patients with oxygen saturation < 95%. Patients were mainly identified at ED admission by shortness of breath (57.6%) during the first wave, whereas they were mostly classified by suspicion of COVID-19 during the second (57.2%) and the third (61.8%) waves. At hospital admission, patients in the first wave were mainly diagnosed with viral pneumonia (64.3%) since the specific code for COVID-19 was not yet available, whereas patients in the second (92.4%) and third (95.4%) waves were confirmed with COVID-19 diagnosis. Of the grouped secondary diagnosis identified at hospital admission, the most prevalent were circulatory system diseases, endocrine, nutritional and metabolic diseases, with essential hypertension and decompensated type 2 diabetes being particularly prevalent (**Table 1**). The number of secondary diagnoses increased with age (**S2 Table**).

## Use of healthcare resources and costs

The mean ± SD LOS for the whole population was 11.2 ± 10.4 days for the first wave, 12.0 ± 12.3 days for the second, and 9.6 ± 6.3 days for the third, representing a decrease of approximately two days between the first two and the third wave.

Regarding the use of procedures, the percentage of COVID-19 patients who had undergone at least one ED procedure was higher during the first (75.8%; n/N = 1667/2198) and second (74.1%; n/N = 488/659) waves than during the third (27.0%; n/N = 180/667). However, each patient received a similar mean ± SD number of procedures throughout the different outbreak waves (1.4 ± 0.6 for the first, 1.2 ± 0.5 for the second, and 1.2 ± 0.4 for the third wave). Likewise, all patients received at least one procedure during hospitalizations, and the mean ± SD number per patient was similar among the different outbreak waves (7.7 ± 3.5 for the first, 9.1 ± 3.5 for the second, and 8.8 ± 3.2 for the third wave) (**S3 and S4 Tables** describe the procedures followed during the ED and the hospital stay).

Concerning laboratory tests, the proportion of patients receiving at least one amounted to around 70% throughout the different waves, although the mean ± SD number of tests per

**Table 1. Summary of patients' demographic and clinical parameters at hospital admission (grouped by outbreak waves).**

| Variables | 1st Wave | 2nd Wave | 3rd Wave |
|---|---|---|---|
| **Age (years), mean ± SD** | 67.9 ± 16.2 | 65.6 ± 16.8 | 65.4 ± 16.6 |
| **Gender (male), n/N (%)** | 1360/2279 (59.7) | 465/740 (62.8) | 449/737 (60.9) |
| **Vital signs at the ED, n/N (%)** | | | |
| Temperature > 38˚C, n/N (%) | 1598/1732 (92.3) | 595/631 (94.3) | 607/628 (96.7) |
| Oxygen saturation < 95%, n/N (%) | 945/1779 (53.1) | 287/618 (46.4) | 333/644 (51.7) |
| Requirement for mechanical ventilation, n/N (%) | | | |
| Oxygen saturation < 95% | 608/945 (64.3) | 171/287 (59.6) | 201/333 (60.4) |
| Oxygen saturation ≥ 95% | 402/834 (48.2) | 165/331 (49.8) | 139/311 (44.7) |
| Heart rate (bpm), mean ± SD | 89.8 ± 16.9 | 88.3 ± 16.3 | 89.1 ± 17.3 |
| Systolic blood pressure (mmHg), mean ± SD | 131.3 ± 22.8 | 132.1 ± 20.6 | 134.1 ± 20.9 |
| Diastolic blood pressure (mmHg), mean ± SD | 74.7 ± 13.0 | 76.6 ± 12.3 | 77.7 ± 13.2 |
| **Diagnosis at the ED, n/N (%)** | | | |
| Common cold | 216/2200 (9.8) | 5/691 (0.7) | 5/707 (0.7) |
| Cough | 139/2200 (6.3) | 14/691 (2.0) | 12/707 (1.7) |
| General discomfort | 73/2200 (3.3) | 20/691 (2.9) | 11/707 (1.6) |
| Fever | 274/2200 (12.5) | 64/691 (9.3) | 26/707 (3.7) |
| Shortness of breath | 1268/2200 (57.6) | 91/691 (13.2) | 114/707 (16.1) |
| Suspicion of COVID 19 | 0/2200 (0) | 395/691 (57.2) | 437/707 (61.8) |
| Another diagnosis | 230/2200 (10.5) | 102/691 (13.0) | 102/707 (14.4) |
| **Primary diagnosis at the hospital admission, n/N (%)** | | | |
| Other viral pneumonia (J12.89) | 1466/2279 (64.3) | 0/740 (0) | 0/737 (0) |
| Pneumonia, unspecified organism (J18.9) | 225/2279 (9.9) | 2/740 (0.3) | 2/737 (0.3) |
| COVID 19 (U07.1) * | 0/2279 (0) | 684/740 (92.4) | 703/737 (95.4) |
| Another diagnosis | 588/2279 (25.8) | 54/740 (7.3) | 32/737 (4.3) |
| **Secondary diagnosis at hospital admission, n/N (%)** | | | |
| Essential (primary) hypertension (I10) | 809/2279 (35.5%) | 251/740 (33.9%) | 260/737 (35.3%) |
| Decompensated type 2 diabetes mellitus (E11.9) | 249/2279 (10.9%) | 72/740 (9.7%) | 64/737 (8.7%) |
| **Grouped secondary diagnosis at hospital admission, d/D (%)** | | | |
| Diseases of the circulatory system (I00-I99) | 2030/11912 (17.0) | 605/3994 (15.1) | 632/3730 (16.9) |
| Certain infectious and parasitic diseases (A00-B99) | 1866/11912 (15.7) | 81/3994 (2.0) | 40/3730 (1.1) |
| Endocrine, nutritional, and metabolic diseases (E00-E89) | 1761/11912 (14.8) | 547/3994 (13.7) | 562/3730 (15.1) |
| Other coronaviruses as a cause of diseases classified under other headings (B97.29) | 1736/11912 (14.6) | 0/3994 (0) | 0/3730 (0) |
| Diseases of the respiratory system (J00-J99) | 1459/11912 (12.2) | 1133/3994 (28.4) | 1142/3730 (30.6) |
| Symptoms, signs and abnormal clinical and laboratory findings, not elsewhere classified (R00-R99) | 844/11912 (7.1) | 232/3994 (5.8) | 208/3730 (5.6) |
| Diseases of the digestive system (K00-K94) | 591/11912 (5.0) | 241/3994 (6.0) | 209/3730 (5.6) |

Abbreviations: COVID-19 (coronavirus disease-2019); d/D where d is the number of secondary diagnoses identified with that code, and D is the number of total secondary diagnoses for our study population; ED (emergency department); SD (standard deviation); *the implementation of the new code for COVID-19 (U07.1) took place in July 2020.

patient was considerably higher for the first wave (129 ± 384) compared with the second (3.6 ± 10.3) and the third (3.8 ± 10.7) waves.

The pharmacological treatment, which was only recorded during the hospital ward stay, also differed between the first wave, where aminoquinolines (87.9%; n/N = 1814/2066) and antivirals for the human immunodeficiency virus (51.8%; n/N = 1070/2066) were generally administered, and the second and the third waves, where glucocorticoids were the most frequently prescribed treatment, administered to 89.9% (n/N = 588/661) of patients in the second

and 91.4% (n/N = 606/667) in the third wave. In addition, during the three outbreak waves, other treatments like heparin, cephalosporins, or anilids were broadly administered to hospitalized patients (details of treatments for the different waves are presented in **S5 Table**).

**Cost analysis for patients not admitted to the ICU (only treated in the general ward).** The mean ± SD LOS for patients who were not admitted to the ICU was similar among outbreak waves: 9.6 ± 6.6 days for the first, 9.7 ± 6.2 days for the second, and 9.0 ± 5.7 days for the third wave. The mean ± SD costs per hospitalized patient amounted to €9895.3 ± €7672.1 for the first wave, €10 196.1 ± €7237.2 for the second, and were slightly lower €9364.5 ± €6321.1 for the third wave. However, when estimated per day, results obtained among waves were comparable: €1050.0 ± €438.5 for the first, €1068.0 ± €342.6 for the second and €1088.6 ± €472.2 for the third wave. Among total costs per patient, hospital LOS incurred the highest cost percentage across outbreak waves and represented approximately 75% of costs, followed by hospital procedures, accounting for 15%, and pharmacological costs, representing around 7% to 9% of total hospital costs (See details in **Table 2** for patients who were not admitted to the ICU).

When total costs per patient were stratified by age groups, they showed a growing trend with older age: the highest costs were observed in patients over 80 (from €10 416.5 to €13 093.3 in the third and second waves, respectively), whereas the lowest corresponded to pediatric patients (from €2102.1 to €4350.0 in the second and third waves, respectively). In addition, mean LOS went from 2.3 to 4 days for pediatric patients (< 12 years) and 10.4 to 12.8 days in patients over 80 (**Fig 3A**. For patients who were not admitted to the ICU). As a result of the GLM model for non-ICU patients, it was found that age and length of stay significantly increased the total cost per patient by 0.06% (95% CI 0.007%-0.11%) and 11.57% (95% CI 11.32%-11.82%) per unit respectively.

However, this trend was not maintained when estimating mean healthcare costs per patient and day, with a mean cost of approximately €1000 for all age groups and waves (**Fig 4A**. For patients who were not admitted to the ICU). **S6 Table** shows mean costs per patient and wave stratified by age and by type of healthcare costs for patients who were not admitted to the ICU.

**Cost analysis for patients admitted to the ICU (treated in the ICU at some point of the inpatient hospital stay).** The mean ± SD LOS for patients who were admitted to the ICU was 26.7 ± 21.6 days for the first, 31.0 ± 26.3 days for the second, and 15.7 ± 8.2 days for the third wave. This represents a difference of approximately nine days between the first two waves and the third wave. Likewise, the mean ± SD cost per hospitalized patient amounted to €77 899,4 ± €63 004.8 and €81 332.5 ± €63 725.8 for the first and second waves, respectively, but decreased to €36 952.1 ± €24 809.2 for the third wave. However, the difference in cost between the first two waves and the third were not as remarkable when estimated per day, in which case they amounted to €3299.4 € ± €3263.6 and €3072.4 ± €2161.0 for the first and second waves, respectively, and €2972.4 ± €3117.9 for the third.

The hospital LOS incurred the largest percentage of costs across outbreak waves and represented around half of the total costs, followed by hospital procedures accounting for more than 30% of costs, and pharmacological costs accounting for 6% to 12% of total hospital costs (See details in **Table 2** for patients who were admitted to the ICU). Likewise, mean healthcare costs per patient also increased with age. However, they reached their highest value in patients aged 60 to 79 (exceeding €100 000 per patient) but decreased by about two thirds for those over 80 (€35 928.9). Mean LOS followed the same pattern with a peak in the second wave for patients aged 60 to 79 (mean LOS over 35 days) and declined by 50% for patients over 80 years of age (mean of 16.1 days) (**Fig 3B**. For patients who were admitted to the ICU). GLM model showed that the third wave vs first wave, as well as sex (female vs male), had a significant impact on the total cost per patient decreasing it with respect to the baseline category by 19.58% (95% CI 5.74%-31.38%) and 12.97% (95% CI 0.47%-23.90%) respectively. On the other

Table 2. Mean and median costs per patient associated with COVID-19 hospitalization in the different outbreak waves stratified by thy type of healthcare costs.

| Population | Waves | N (%) | Type of healthcare costs | | | | | | | | | | TOTAL COSTS PER PATIENT |
| | | | Hospital procedures | | Emergency procedures | | Laboratory test | | Pharmacological treatment | | Hospital LOS | | |
| | | | Mean (median) costs (€) | Percentage of total costs (%) | Mean (median) costs (€) | Percentage of total costs (%) | Mean (median) costs (€) | Percentage of total costs (%) | Mean (median) costs (€) | Percentage of total costs (%) | Mean (median) costs (€) | Percentage of total costs (%) | Mean (median) costs (€) |
|---|---|---|---|---|---|---|---|---|---|---|---|---|---|
| Patients not admitted to the ICU | 1st Wave | 2066 (90.7%) | 1441 (734) | 14.6 | 63 (22) | 0.6 | 241 (93) | 2.4 | 602 (497) | 6.1 | 7 547 (6 300) | 76.3 | 9895 (7 715) |
| | 2nd Wave | 661 (89.3%) | 1472 (1014) | 14.4 | 54 (22) | 0.5 | 106 (93) | 1.0 | 902 (224) | 8.9 | 7 661 (6 300) | 75.1 | 10 196 (8 529) |
| | 3rd Wave | 667 (90.5%) | 1477 (1046) | 15.8 | 21 (0) | 0.2 | 104 (93) | 1.1 | 696 (1634) | 7.4 | 7 066 (6 300) | 75.5 | 9364 (7 920) |
| Patients admitted to the ICU | 1st Wave | 213 (9.3%) | 28 400 (37 033) | 36.5 | 53 (22) | 0.1 | 1931 (302.6) | 2.5 | 8 070 (4 370) | 10.4 | 39 445 (27 366) | 50.6 | 77 899 (67 343) |
| | 2nd Wave | 79 (10.7%) | 27 181 (22 848) | 33.4 | 29 (22) | 0.0 | 238 (186) | 0.3 | 9 766 (4 834) | 12.0 | 44 117 (31 277) | 54.2 | 81 332 (63 170) |
| | 3rd Wave | 70 (9.5%) | 14 155 (5 215) | 38.3 | 12 (0) | 0.0 | 154 (139) | 0.4 | 2 375 (1 803) | 6.4 | 20 255 (18 085) | 54.8 | 36 952 (27 930) |

Abbreviations: ICU (intensive care unit); LOS (length of stay)

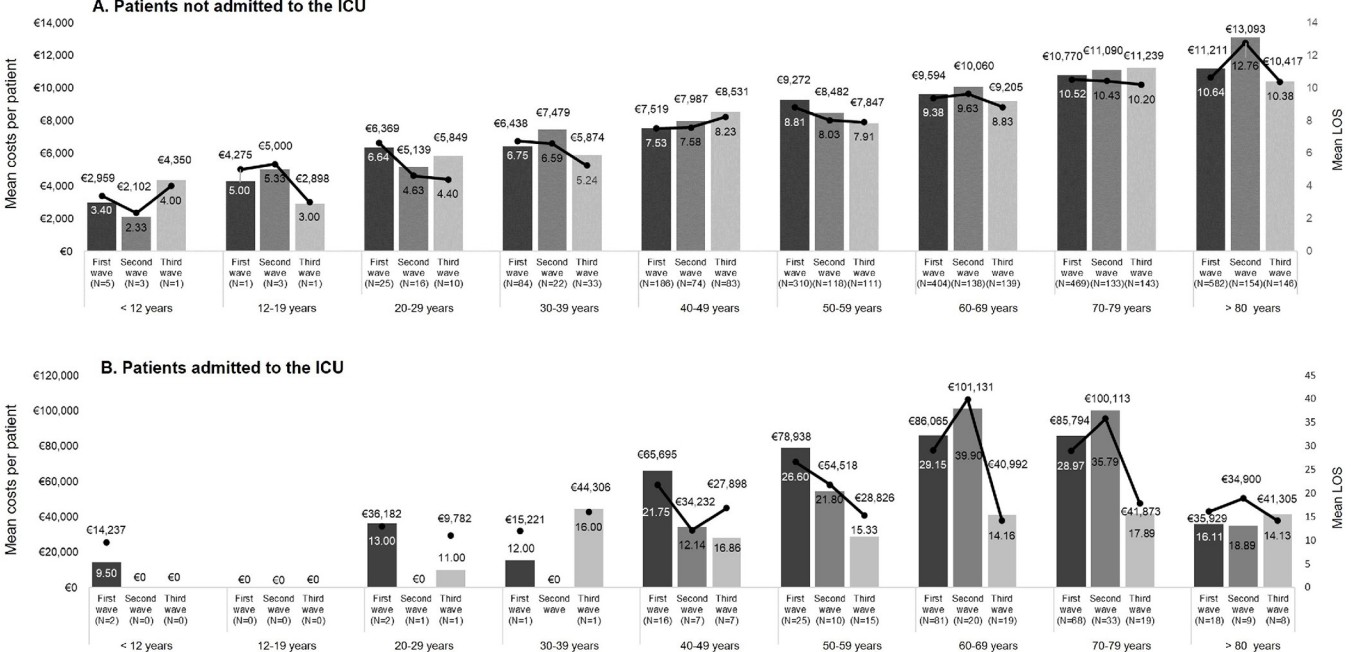

**Fig 3.** Mean costs and LOS per patient associated with COVID-19 hospitalization in the different outbreak waves stratified by the vaccination age bands for patients who were not (A) and who were (B) admitted to the ICU. Abbreviations: ICU (intensive care unit); LOS (length of stay). Costs are estimated as unadjusted means.

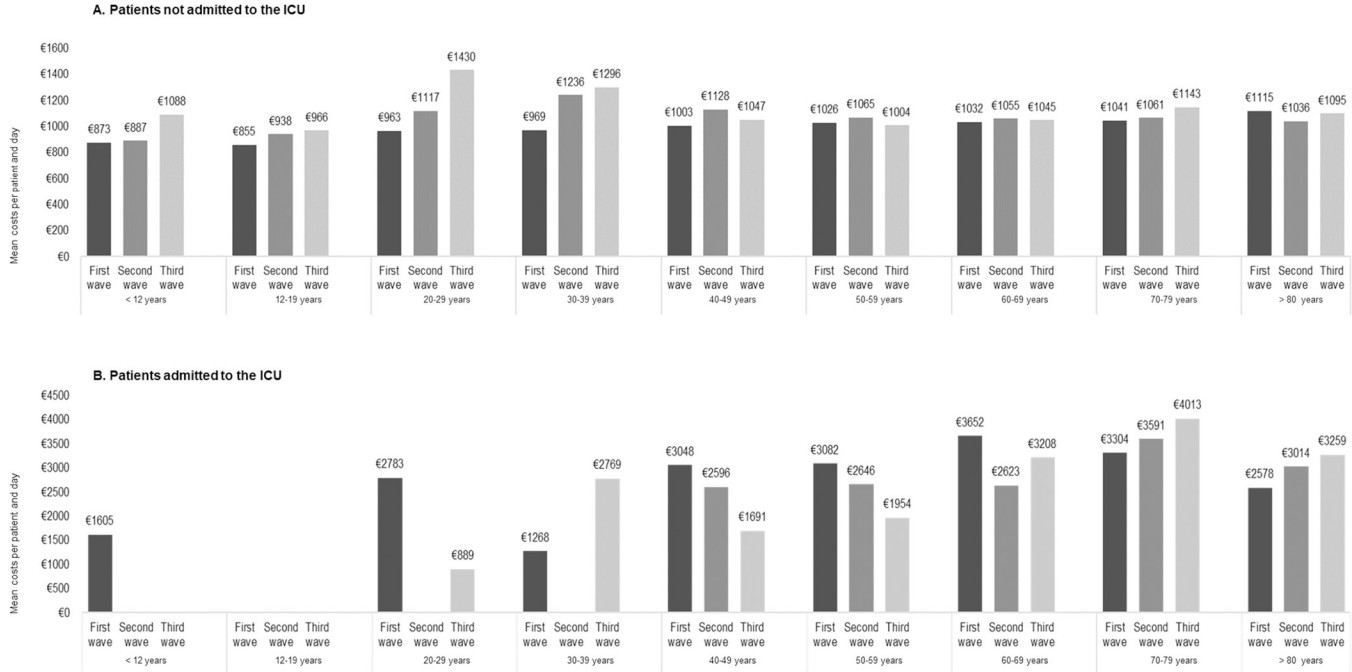

**Fig 4.** Mean costs per patient and day associated with COVID-19 hospitalization in the different outbreak waves stratified by the vaccination age bands for patients who were not (A) and who were (B) admitted to the ICU. Abbreviations: ICU (intensive care unit). Costs are estimated as unadjusted means.

hand, in this group, the length of stay significantly increased the total cost per patient by 3.48% (95% CI 3.05%-3.90%) per unit.

When estimating the mean cost per patient and day, differences among age groups were reduced and varied across waves from €1691 to €4013 for patients over 40 (**Fig 4B**. For patients admitted to the ICU). **S7 Table** shows mean costs per patient and wave stratified by the age and by the type of healthcare costs for patients who were admitted to the ICU.

## Sensitivity analysis with two alternative scenarios

Results from the sensitivity analysis were consistent across the populations evaluated, outbreak waves and age groups: the analysis with the scenario including the minimum individual costs from laboratory tests and hospital stay resulted in lower mean costs; whereas the scenario with the maximum costs generated higher mean costs across all the subgroups evaluated. (See the details of sensitivity results in **S1 Text** and **S8 Table**).

## Discussion

We have described the use of healthcare resources and costs in a cohort of hospitalized COVID-19 patients in Spain and reported trends across the different outbreak waves during the first year of the pandemic. We retrieved the data from the HM Hospitals consortium's centralized database: "COVID Data Saves Lives". This database registered data on the COVID-19 patients who were attended in any of these private hospitals across different Spanish regions during the main peaks of infection in Spain: from February 2020 until the mass vaccination campaigns at the beginning of 2021 [20].

Most patients in our study received care exclusively in the hospital ward (about 90%); for this population, the resulting costs per patient were similar across outbreak waves (from a mean ± SD of €9364.5 ± €6321.1 in the third wave to €10 196.1 ± €7237.2 in the second). In addition, our mean values were within those obtained in other countries: €1877.5 to €14 232.8 per hospitalized patient [14–18] but lower than that reported for a Spanish cohort by Carrera-Hueso et al. [19] including patients who were not admitted to the ICU (€50 132). In the aforementioned study, higher costs were attributed to a longer mean LOS than that observed in our study (mean of 44.1 days vs 9 days in our study). Additionally, unlike ours, their analysis included other direct medical costs such as medical visits and nursing hours, which may also have contributed to the higher per-patient costs.

In our study, patients who required intensive care at some point of their hospital stay represented around 10% of our study population. As expected, they incurred considerably higher costs than those who were not admitted to the ICU—costs per hospitalized patient were up to eight-fold higher in this population. Furthermore, the increase in costs was linked to a considerably longer LOS, up to three times longer than for patients not admitted to the ICU. In agreement with our findings, prior research found a considerable difference between the group of patients requiring ICU admission and those that did not require intensive care. In this respect, the average costs were between 4 [19, 30] and 6 times [16] higher in patients requiring ICU admission than in those who did not. Another noteworthy point regarding patients requiring intensive care was that their mean LOS, and consequently mean costs, decreased by approximately half in the third wave. This decline in LOS in the third wave might be due to various factors. On the one hand, the improvement in the management of critical patients during the first months of the pandemic led to faster patient recovery. And, on the other hand, the introduction of rapid antigen diagnostic tests in the third wave, which allowed early detection and treatment of critical COVID-19 patients [31] and, therefore, contributed to an earlier discharge from hospital.

We should also point out that mean costs per patient and per day in our study cohort did not differ substantially between the outbreak waves and were similar to those obtained in another analysis conducted at a private hospital in Spain, where mean costs per patient and per day were €875.6 for general ward stay, and €2486.2 for ICU stay [32].

However, although the costs were similar among the waves in our study, the use of some hospital resources changed dramatically; for example, the percentage of patients who underwent an ED procedure decreased by approximately 64% in the third wave compared to the first and second waves (from 75.8% in the first, 74.1% in the second to 27.0% in the third wave). Also, the mean number of laboratory tests per patient declined greatly after the first wave (from a mean of 129 tests per patient in the first to 3.6 in the second and 3.8 in the third wave). This substantial decrease in certain procedures might also be attributed to the continuous learning process during the first months of the pandemic and the rapid adoption of clinical guidelines for COVID-19 patient management [33–35]. The fact that the total costs per patient were not affected by this decrease might be because laboratory tests only represented a small percentage of the total healthcare costs in our study (i.e., 2.4% for patients not admitted to the ICU and 2.5% for those admitted to the ICU in the first wave), with hospital admission representing the largest share of the costs.

In addition, the treatments administered to patients differed drastically between the first wave and the other two. In this respect, during the first wave, most patients in our sample were treated with aminoquinolines and antivirals, whereas in the second and third waves, patients were treated mainly with corticosteroids. This shift in treatment type in favor of corticosteroids was also found in other studies in Spain [9, 10, 36, 37] and other settings [38] and might also reflect the rapid incorporation of clinical evidence generated during this short period [35].

Another interesting observation was that mean costs per COVID-19 patient and, especially, the LOS increased with age in our study cohort. Mean LOS increased approximately four-fold for the population over 80 compared to pediatric patients (< 12 years) (from a mean of 2.3 to 4.0 days per patient < 12 years to a mean of 10.4 to 12.8 days for patients >80). The relationship between age and hospital costs has been observed in previous studies, reporting older age as one of the major drivers of costs and hospital LOS [14, 30]. The results of the regression model confirm that length of stay is one of the main factors influencing cost in both ICU and non-ICU populations of our study sample. In the non-ICU population, age was also an important driver. The fact that older patients incurred higher costs and LOS than younger ones can be explained, as older patients are more likely to suffer certain comorbidities such as hypertension, diabetes, or cardiovascular diseases, which increase the risk of severe disease [39]. This relationship between older patients with comorbidities was also observed in our study, where the number of secondary diagnoses, such as circulatory systemic or endocrine diseases, increased with the patient's age. Furthermore, older patients suffer from declining immunity, making them more vulnerable to developing severe or critical COVID-19 [40]. Both comorbidities and diminished immunity increase the need for admission to intensive care and longer LOS, thus considerably raising healthcare costs [14, 30].

Nonetheless, the fact that older patients might be more prone to developing severe or critical COVID-19 does not always involve higher hospital costs. In fact, among the patients admitted to the ICU in our study, patients over 80 incurred lower mean costs and LOS per patient than the 50- to 79-year-old age groups. A cost analysis by Tsai et al. [15] observed that, although patients aged 75 or older were more likely to be hospitalized, their hospitalizations involved lower costs than younger patients. The authors hypothesized that lower costs in older patients might be attributed to worse prognosis and higher mortality rates, resulting in shorter inpatient stays.

## Limitations

Our cost analysis has some limitations that should be mentioned. First, the data were extracted from the HM Hospitals database, which agglutinated data from a consortium of private hospitals in Spain. Furthermore, this sample of hospitals represented 2.2% of Spanish hospitals [41]. In that respect, the study population might not accurately represent the population attended in the Spanish healthcare system. Nonetheless, we consider that our results could be reasonably representative of the Spanish healthcare system at the population and cost-analysis level. At the population level, unlike other studies that represented only one area or autonomous region [9, 10, 19, 42], our database included data from COVID-19 patients in 17 hospital catchment areas across four regions in Spain (Autonomous Community of Madrid, Castilla and Leon, Catalonia, and Galicia), covering an extensive and diverse demographic population. In addition to the extensive geographic area, HM Hospitals put their facilities at the public administration's disposal and absorbed COVID-19 patients from public hospitals during the outbreaks, treating more than 40,000 COVID-19 patients during 2020 [43, 44]. We believe that the main differences between the public and private hospitals prior to the pandemic may be precisely in the length of stay, however, the average length of stay is highly dependent on the pathology [45], which management was equally unknown in both settings. It is important to note that previous studies carried out in Spain in public hospitals during the first wave showed the average length of stay figures similar to those described in our study [46, 47]. At a cost-analysis level, the unitary costs corresponding to procedures, laboratory tests or hospital stays were obtained from official sources at a national level or from the different regions. Therefore, we suggest that the costs obtained are an approximation of the actual costs of the Spanish public healthcare system.

Secondly, data on the pharmacological treatments used were not available for the ICU stays; consequently, costs for patients admitted to the ICU at some point of their stay might be underestimated. Additionally, as we recorded pharmaceutical prices from official sources (selling price), we might have ignored the actual subsidized prices applied in public hospitals. Thirdly, all the treatments and procedures during hospitalization were recorded regardless of whether they were aimed at treating COVID-19 or not. Similarly, secondary diagnoses were recorded regardless of whether they were comorbidities or symptoms and consequences of COVID-19 itself. As a result, costs could have been overestimated for older patients or patients with various comorbidities or more symptoms due to COVID-19. However, we believe our results reflect the real impact of this disease, which affects older patients with different comorbidities the most. For this reason, we deemed it appropriate to consider all costs together, as they better reflect patients' health status and care needs. Finally, only healthcare costs related to hospital admission and the use of medical resources were considered in the cost analysis, whereas other costs such as those related to healthcare workers were not considered, thus we could have underestimated the total costs. We also acknowledge that the non-inclusion of other costs, such as those related to the loss of productivity might also underestimate the costs in working-age patients, especially those aged between 40 and 60, who account for about one-fourth of the population (from 23.6% to 28.2%).

## Conclusions

This study describes the use of healthcare resources and costs in a cohort of hospitalized COVID-19 patients over the course of the first year of the pandemic. Accordingly, our main observation is that mean costs per hospitalization COVID-19 patient did not consistently differ across the three main outbreak waves in our sample, especially for those patients who were not admitted to the ICU. However, longer LOS in patients admitted to the ICU (especially in

the first two waves) and in older patients were the most important drivers of hospitalization costs. Also, the use of resources changed across outbreak waves, with a decrease in hospital procedures and laboratory tests, and an increase in the use of corticosteroids, reflecting the rapid incorporation of clinical evidence into routines during this short period.

In summary, our cost analysis provides robust data that could be useful to inform decision-makers about how health systems were impacted across the different outbreak waves in the first year of the COVID-19 pandemic.

## Supporting information

**S1 Fig. Distribution of patients according to vaccination age bands in the different outbreak waves.**
(TIF)

**S1 Table. STROBE checklist.**
(DOC)

**S2 Table. Distribution of number of secondary diagnoses according to age vaccinations.**
(DOC)

**S3 Table. Distribution of procedures during ED stay in the different outbreak waves.** All patients.
(DOC)

**S4 Table. Distribution of procedures during hospital stay in the different outbreak waves.** All patients.
(DOC)

**S5 Table. Proportion of patients receiving different pharmacological treatments in the different outbreak waves.** Recorded only during the hospital ward stay.
(DOC)

**S6 Table. Mean costs per patient associated with COVID-19 hospitalization in the different outbreak waves stratified by the vaccination age bands and by the type of healthcare costs.** Patients who were not admitted to the ICU.
(DOC)

**S7 Table. Mean costs per patient associated with COVID-19 hospitalization in the different outbreak waves stratified by the vaccination age bands and by the type of healthcare costs.** Patients admitted to the ICU.
(DOC)

**S8 Table. Results from the sensitivity analysis: Mean costs per patient associated with COVID-19 hospitalization in the base case and in the alternative scenarios (minimum costs and maximum costs) in the different outbreak waves.**
(DOC)

**S1 Text. Detailed results from the sensitivity analysis.**
(DOC)

## Acknowledgments

The authors want to thank Lucía Pérez-Carbonell at Outcomes'10 (Castellón, Spain) for their assistance with medical writing.

## Author Contributions

**Conceptualization:** Georgina Drago, Francisco Javier Pérez-Sádaba, Susana Aceituno, Carla Gari, Juan Luis López-Belmonte.

**Data curation:** Francisco Javier Pérez-Sádaba, Susana Aceituno, Carla Gari.

**Formal analysis:** Carla Gari.

**Funding acquisition:** Georgina Drago, Juan Luis López-Belmonte.

**Investigation:** Francisco Javier Pérez-Sádaba, Susana Aceituno, Carla Gari.

**Methodology:** Francisco Javier Pérez-Sádaba, Susana Aceituno, Carla Gari.

**Project administration:** Georgina Drago, Francisco Javier Pérez-Sádaba, Susana Aceituno, Juan Luis López-Belmonte.

**Software:** Carla Gari.

**Supervision:** Georgina Drago, Francisco Javier Pérez-Sádaba, Susana Aceituno, Juan Luis López-Belmonte.

**Validation:** Georgina Drago, Francisco Javier Pérez-Sádaba, Susana Aceituno, Carla Gari, Juan Luis López-Belmonte.

**Visualization:** Georgina Drago, Francisco Javier Pérez-Sádaba, Susana Aceituno, Carla Gari, Juan Luis López-Belmonte.

**Writing – original draft:** Georgina Drago, Francisco Javier Pérez-Sádaba, Susana Aceituno, Carla Gari, Juan Luis López-Belmonte.

**Writing – review & editing:** Georgina Drago, Francisco Javier Pérez-Sádaba, Susana Aceituno, Carla Gari, Juan Luis López-Belmonte.

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
