## [Decision Letter · Decision Letter 0]

18 Jul 2022

PONE-D-22-09612Healthcare resource use and associated costs of hospitalized COVID-19 patients in Spain: a retrospective analysis from the first to the third pandemic wave. EPICOV study.PLOS ONE

Dear Dr. Drago,

Thank you for submitting your manuscript to PLOS ONE. After careful consideration, we feel that it has merit but does not fully meet PLOS ONE’s publication criteria as it currently stands. Therefore, we invite you to submit a revised version of the manuscript that addresses the points raised during the review process.

We look forward to receiving your revised manuscript.

Kind regards,

Martial L Ndeffo Mbah, Ph.D

Academic Editor

PLOS ONE

Journal Requirements:

2. For certain types of papers, “consent to publish” is required, and this may apply to both images and data/text. This is an additional, and different, type of consent to that required to participate in the study (ie. is independent of IRB oversight). Consent to publish is needed for either/both, 

a) images that identify participants/patients in a study (ie photos of patients or people); 

b) detailed descriptions of individual participants/patients (in particular their clinical histories) that are of sufficient detail that it is possible for a reader to identify the specific person/people.

PLOS policy also states “Authors must also take special care when submitting manuscripts that contain potentially identifying images of people. Identifying information should not be included in the manuscript unless the information is crucial”. 

This issue does not apply for papers that use “photograph libraries” or “photograph databases” (often used in psychology studies) and which typically show thumbnails of individuals from those libraries (often as examples of the experimental setup). In these cases the authors will normally name an image database and give a URL for the database/library. For images from photo libraries or photo databases, see the desk notes for Copyright.

Additional Editor Comments (if provided):

Thank you very much for your patient with this usually long review process. Based on the reviewers comments, we would like to invite you to submit a revised version of your manuscript which addresses all of the reviewers concerns. It is our perception that these comments will help improve the quality of our manuscript and make it more suitable for publication.

Reviewers' comments:

Reviewer's Responses to Questions

**Comments to the Author**

1. Is the manuscript technically sound, and do the data support the conclusions?

Reviewer #1: Yes

Reviewer #2: Partly

2. Has the statistical analysis been performed appropriately and rigorously? 

Reviewer #1: No

Reviewer #2: No

3. Have the authors made all data underlying the findings in their manuscript fully available?

Reviewer #1: No

Reviewer #2: No

4. Is the manuscript presented in an intelligible fashion and written in standard English?

Reviewer #1: Yes

Reviewer #2: Yes

5. Review Comments to the Author

Reviewer #1: Thank you for the opportunity to review this manuscript on healthcare utilization and cost of COVID-19. While the information on cost of COVID-19 care is important, I believe it is important that it includes adjusted measures of association, in order to answer the questions asked in the paper.

Page 5, line 97: Please give some insight into how representative these hospitals are of the whole Spain population. What % of total hospitals do they amount to? Are patient or hospital characteristics different? For example, are private hospitals characterized by better standards of care, compared to public ones? It is important in understanding who is represented in this study.

Page 5, line 107: Suggest making this sentence more concise and list the specific date ranges here right away, supported by citations. It does not seem that holiday information (eg Christmas) adds much important information to this.

Page 6, line 114: Please provide more information on how the anonymized data were collected. What was the response rate? Are all patients with COVID-19 in these hospitals included?

Page 7, line 134: Were patients exclusively admitted through the ED? If so please specify in the text.

Page 7: Was information available on underlying medical conditions?

Page 8: Were any statistical models used? It seems that GLM models with Gamma distribution accounting for specific patient and/or hospital characteristics are needed to compare the costs by these characteristics. All models should be adjusted for age, sex, race/ethnicity, underlying medical conditions (if available) With unadjusted measures of association, it is hard to say whether some of these subpopulations or waves are actually different or whether the difference is driven by other characteristics. For example, the statement in the Conclusions section states that age and ICU admission were the most important drivers of the cost; but it is not possible to assess that without adjusting for all these factors in one model.

Page 8: Means in this skewed distribution are very much affected by the tail length and outliers; suggest using median costs, in addition to mean costs.

Page 18: The “increase” of LOS and costs with age represents unadjusted means (is that correct?). Suggest using GLM models with age group as the covariate of interest (adjusted for other important controls) to estimate whether this difference was indeed meaningful and significant.

Methods: Suggest plotting the cost distribution for full sample and/or certain subsets – so that we can understand the skewness and min/max costs.

Methods: Please consider a supplemental analysis that would censor the costs above 99th percentile. It is possible that means may not be so different after removing those super-utilizers with extremely high costs.

Methods and Results: While mean costs are interesting, it would also be interesting to understand the specific contributions of all patient characteristics to cost in terms of percent. For example, as age increases, what is the increase in costs in percent? This gives us some understanding of the magnitude of this difference (2% or 50%), rather than just dollar terms.

Figures: Please add footnotes describing what is shown in each figure (whether these are adjusted or unadjusted cost means). Each figure must stand alone and be interpretable without looking at the Methods.

Reviewer #2: Major issues:

(1) There is already a paper which present costs related to cost of COVID cases. The authors cited it https://healtheconomicsreview.biomedcentral.com/articles/10.1186/s13561-021-00340-0

Then what is the added value of the current paper?

(2) In my view it is not ok to mix cost and mortality. Your paper deals mainly with cost and there are a lot of information that you can present about costs. I recommend to make a different paper in mortality/survival, etc.

(3) The paper has rather a descriptive manner, rather a case-study. If the sample is random, I suggest to make an inference of COVID cost’s

(4) Related to (1),(3) . Probably is more important to see what combination of factors/covariates drive to higher costs? This can be an added value of the paper. In this sense, I suggest to use for example regression modelling.

(5) You present some results about sensitivity, but in the methods the sensitivity analysis is described too vague. It is DeterministicSA or ProbabilitySA, I guess is DSA.

(6) You deal with samples in each wave, why you do not test if the difference is statistically significant? Between waves and or between groups of patients.

(7) We know that LOS and age are determinants for higher costs, but it is something else a characteristic of COVID hospitalized cases? If you want to add a value to your study, why not a comparison with severe classic Pneumonia cases? If the cost are not significantly higher, than….

(8) In the sense of (7) I suggest to search for a combination of factors such as , age-group+ group of comorbidities.

(9) Can we see a distribution of patients by cost? I believe is a long tail distribution, then the mean cost may be not relevant….Eg. you sum 3patients*2000 +1patients *10000 and you have a mean of 4000. The range of cost is not enough relevant.

(10) Maybe a synthetic 8 figure/diagram is useful to understand the patient path-way. Eg diagnosed at family medicine-� covid ward-� ICU, or Non-diagnosed----emergency---ICU.

(11) For the hospital costs we have a DRG cost? Then, the structure of patients per hospital is the same in each wave? I think we have a hospital or regional effect cost when DRG is computed. Please clarify!

Minor issues

(1) Reference [27] has a broken link

(2) If you decide to keep mortality in the analysis then , in the methodology some description, formula or something else should appear about mortality. Then if then, maybe you should take into account

(3) I don’t know if it so important to use so many age-groups . The difference are small. Using so many age-groups you reduce sub-sample size. Why not children? (<18), Then 18-29, 30-49, 50-69, 70+, or something similar.

(4) Secondary diagnosis at hospital admission, n/N (%) please use other notation, Usually n= the sample size and N is the population size.

(5) Secondary diagnosis at hospital admission, n/N (%). I don’t understand the shares. In my view we should see how many COVID patients have an I10 or E11 as secondary diagnostic. Now I think it is a share of how many I10 are in the secondary diagnostics divided by patients*nr_of secondary diagnostics ?!. Please clarify!

6. PLOS authors have the option to publish the peer review history of their article (what does this mean?). If published, this will include your full peer review and any attached files.

Reviewer #1: No

Reviewer #2: No

---

## [Author Response · Author response to Decision Letter 0]

21 Oct 2022

Because the response to the reviewers includes tables and graphs, we have included it in one of the documents attached to the manuscript. See Response to reviewers_V.01_SAN

---

## [Decision Letter · Decision Letter 1]

21 Nov 2022

PONE-D-22-09612R1Healthcare resource use and associated costs of hospitalized COVID-19 patients in Spain: a retrospective analysis from the first to the third pandemic wave. EPICOV study.PLOS ONE

Dear Dr. Drago,

Thank you for submitting your manuscript to PLOS ONE. After careful consideration, we feel that it has merit but does not fully meet PLOS ONE’s publication criteria as it currently stands. Therefore, we invite you to submit a revised version of the manuscript that addresses the points raised during the review process.

We look forward to receiving your revised manuscript.

Kind regards,

Martial L Ndeffo Mbah, Ph.D

Academic Editor

PLOS ONE

Journal Requirements:

Reviewers' comments:

Reviewer's Responses to Questions

**Comments to the Author**

1. If the authors have adequately addressed your comments raised in a previous round of review and you feel that this manuscript is now acceptable for publication, you may indicate that here to bypass the “Comments to the Author” section, enter your conflict of interest statement in the “Confidential to Editor” section, and submit your "Accept" recommendation.

Reviewer #1: (No Response)

Reviewer #2: (No Response)

2. Is the manuscript technically sound, and do the data support the conclusions?

Reviewer #1: Partly

Reviewer #2: No

3. Has the statistical analysis been performed appropriately and rigorously? 

Reviewer #1: No

Reviewer #2: No

4. Have the authors made all data underlying the findings in their manuscript fully available?

Reviewer #1: No

Reviewer #2: Yes

5. Is the manuscript presented in an intelligible fashion and written in standard English?

Reviewer #1: Yes

Reviewer #2: Yes

6. Review Comments to the Author

Reviewer #1: I appreciate the authors’ responses to my comments. While most of my comments and questions have been resolved, I have two that have not been fully resolved. My focus is especially on the adjusted measures of association; although authors have provided them, I feel that the table/figure needs to be included in the manuscript and explained in more detail as well.

R1, Comment 1: I feel that my comment about the representativeness was only partially addressed. Could the authors add in the text the total number of hospitals in Spain that were open during the pandemic, as well as the % that the 17 hospitals included in the study represent? I appreciate the added limitation, but quantifying the representativeness would be helpful here.

R1, Comment 6: I appreciate that the authors addressed my comment by using adjusted GLM models and providing adjusted measures of association in the text. I have some more comments on the modeling, however.

. More comments below:

- Estimates need to include confidence intervals.

- Why aren’t all estimates shown in the table in the authors’ responses? I see only age and LOS as predictors of cost among non-ICU patients and wave (but only third wave vs first wave), sex, and LOS as a predict among ICU patients. I would suggest that all coefficients are presented.

- Even more important than coefficients, predicted costs from the GLM models can be used in the place of unadjusted mean costs in the figures. Their 95% Cis can show whether the costs in the three waves were truly different.

- Are these estimates currently included in the main figures? If estimates are not included in the main results, I would suggest replacing one of the figures with the adjusted predicted costs (and 95% CI) in each wave. That can be easily done by using the “margins” Stata post-estimation command to get predicted costs. This would help us see if the costs were truly different by wave after controlling for other covariates.

- To estimate differences by wave for each characteristic (Figure 3A), authors may want to include both wave categorical variable and its interaction with those characteristics (eg wave (1,2,3), age (<12, 12-19, 20-29, 30-39, 40-49, 50-59, 60-69, 70-79, 80), and wave*age). They can then use the Stata “margins” post-estimation command to predict the cost in each wave and at each level of covariate included in the interaction.

Reviewer #2: I appreciate the work of authors but there are some issue which in my view do not match.

The major thing is that the title, the abstract and the conclusions suggest that achieved results characterize the cost/patient for entire population of COVID-19 patient in Spain. The sample selection and thus the methods and the results do not support this hypothesis. In my view there two options (1) if the authors try to extend the costs results for a typical ICU/non-ICU patient from Spain, then they need a representative sample; (2) if the authors cannot achieve a representative sample, I recommand to underline the fact, that this is work is just a case-study, and everywhere in the paper should be clearly specified that "in our/current sample"/ "in our/current case", the results are. The phrases which induce the extension of results should be avoided in my view. Indeed in the limitations paragraph the authors try to explain differences between the current sample and the population, but the arguments are not solid. My worries are related to the thing that costs may be overestimated in particular hospitals compared with anothers. There is no proof that error are compensating each-other.

7. PLOS authors have the option to publish the peer review history of their article (what does this mean?). If published, this will include your full peer review and any attached files.

Reviewer #1: No

Reviewer #2: No

---

## [Author Response · Author response to Decision Letter 1]

16 Dec 2022

Reviewer #1: 

I appreciate the authors’ responses to my comments. While most of my comments and questions have been resolved, I have two that have not been fully resolved. My focus is especially on the adjusted measures of association; although authors have provided them, I feel that the table/figure needs to be included in the manuscript and explained in more detail as well.

Response to reviewer #1:

We appreciate very much the comments received towards our work. Find below the detailed responses to each of the proposed questions.

R1, Comment 1: I feel that my comment about the representativeness was only partially addressed. Could the authors add in the text the total number of hospitals in Spain that were open during the pandemic, as well as the % that the 17 hospitals included in the study represent? I appreciate the added limitation, but quantifying the representativeness would be helpful here.

We welcome suggestions in this regard. Even though the hospitals included are only a small sample of hospitals in the Spanish health system (2.2% of all Spanish hospitals, public and private), we suggest that the patients included in this study could represent the reality of the health system during the first waves of the pandemic. However, the reader could interpret this as inferential statistics, which is not our objective, but rather describe what happens in our study population. For this reason, we wanted to clarify and highlight at some points in the manuscript that our findings refer to the studied sample.

In this regard, we have introduced the following changes in different sections of the manuscript: 

- Title of the manuscript to: “Healthcare resource use and associated costs in a cohort of hospitalized COVID-19 patients in Spain: a retrospective analysis from the first to the third pandemic wave. EPICOV study.”

- Conclusions section from the Abstract (line 39): “LOS was longer for patients admitted to the ICU (especially in the first two waves) and for older patients in our study cohort; these populations incurred the highest hospitalization costs.”

- Introduction section (lines 81-82): “In the present study, we aimed to use real-world data to describe the use of healthcare resources and the associated costs in a cohort of hospitalized patients due to SARS-CoV-2 during three different outbreak waves from the perspective of Spain's national health system (NHS).”

- Discussion section:

Lines 396-396: “We have described the use of healthcare resources and costs in a cohort of hospitalized COVID-19 patients in Spain”

Line 432: “We should also point out that mean costs per patient and per day in our study cohort did not differ substantially between the outbreak waves”

Lines 449-450: “In addition, the treatments administered to patients differed drastically between the first wave and the other two. In this respect, during the first wave, most patients in our sample were treated with aminoquinolines and antivirals”

Lines 458-459: “Another interesting observation was that mean costs per COVID-19 patient and, especially, the LOS increased with age in our study cohort.”

Line 464: “The results of the regression model confirm that length of stay is one of the main factors influencing cost in both ICU and non-ICU populations of our study sample.”

Lines 486-492: “Our cost analysis has some limitations that should be mentioned. First, the data were extracted from the HM Hospitals database, which agglutinated data from a consortium of private hospitals in Spain. Furthermore, this sample of hospitals represented 2.2% of Spanish hospitals (41). In that respect, the study population might not accurately represent the population attended in the Spanish healthcare system. Nonetheless, we consider that our results could be reasonably representative of the Spanish healthcare system at the population and cost-analysis level.”

A reference has been included to support the included text: Hospital statistics - National tables (2020). [Cited 7 December 2022]. In: Spanish Ministry of Health [Internet]. Available from: https://www.sanidad.gob.es/estadEstudios/estadisticas/docs/TablasSIAE2020/Tablas_Nacionales_2020.pdf

Regarding this point, we would like to point out that a sensitivity analysis was carried out taking into account a possible under- or overestimation of the cost.

Lines 506-508: “Therefore, we suggest that the costs obtained are an approximation of the actual costs of the Spanish public healthcare system.”

- Conclusions section:

Lines 530-534: “This study describes the use of healthcare resources and costs in a cohort of hospitalized COVID-19 patients over the course of the first year of the pandemic. Accordingly, our main observation is that mean costs per hospitalization COVID-19 patient did not consistently differ across the three main outbreak waves in our sample, especially for those patients who were not admitted to the ICU”

Lines 540-541: “In summary, our cost analysis provides robust data that could be useful to inform decision-makers about how health systems were impacted across the different outbreak waves in the first year of the COVID-19 pandemic.”

R1, Comment 6: I appreciate that the authors addressed my comment by using adjusted GLM models and providing adjusted measures of association in the text. I have some more comments on the modeling, however.

More comments below:

- Estimates need to include confidence intervals.

Thank the reviewer for this comment. We have added the confidence intervals in the manuscript in section Results, lines 333-334 and lines 378-380.

To facilitate the interpretation of the results obtained, they have been included in the form of percentages, so that the explanation of the coefficients obtained is more intuitive for the reader.

Variable Coef. (exponentiated form) Std. Error p-value IC 95%

 Non ICU-patients

Age 1.000593 0.000265 0.025 1.000074 1.001113

Length of stay 1.115735 0.0012762 0.000 1.113236 1.118239

 ICU patients

Third wave vs fisrt wave 0.8042193 0.0651603 0.007 0.6861319 0.9426303

Second wave vs fisrt wave 1.00022 0.0769607 0.998 0.8602026 1.163028

Sex (Female vs male) 0.8703277 0.0595943 0.043 0.7610236 0.995331

Length of stay 1.034786 0.0021719 0.000 1.030538 1.039051

- Why aren’t all estimates shown in the table in the authors’ responses? I see only age and LOS as predictors of cost among non-ICU patients and wave (but only third wave vs first wave), sex, and LOS as a predict among ICU patients. I would suggest that all coefficients are presented.

A stepwise GLM model was used to carry out the estimation; this type of modelling eliminates those factors that are not significant from the estimation. For predictors whose p-value is greater than 0.05, no coefficient is obtained.

In the case of ICU patients, only the third wave vs. the first wave was significant. However as a categorical variable, the coefficient for the second wave vs. the first wave was estimated too, as it should be included in the modelling.

We have added an additional more detailed explanation of Stepwise in the manuscript in section Analysis, lines 215-219:

“To identify cost drivers of the covariates in the study population a Generalized Linear Model (GLM) with gamma distribution and log link with stepwise algorithm was used. The stepwise regression consists of iteratively adding and removing predictors, in the predictive model, in order to find the subset of variables in the data set resulting in the best performing model, that is a model that lowers prediction error.”

- Even more important than coefficients, predicted costs from the GLM models can be used in the place of unadjusted mean costs in the figures. Their 95% Cis can show whether the costs in the three waves were truly different.

We are very grateful to the reviewer for this and the following two comments, we consider that including the GLM model in the publication has helped us to compare costs according to different population characteristics and has reinforced the conclusions that a priori were drawn from a purely descriptive analysis. However, modifying the mean costs obtained by post-estimation of these through the coefficients, we believe that it modifies the descriptive objective of the project and would imply completely restructuring the publication. The aim of this analysis is not to model the results and thus smooth out heterogeneity, but to evaluate and describe the results obtained directly by the patient, so we consider that presenting the median helped to present greater detail in this respect. Even though this estimation would be of real interest, it may require a refocusing of the project. This would not only include additional analyses but replace the analysis that we have carried out as the aim of the study. Moreover, including all this analysis would overextend and complicate the interpretation of the results.

- Are these estimates currently included in the main figures? If estimates are not included in the main results, I would suggest replacing one of the figures with the adjusted predicted costs (and 95% CI) in each wave. That can be easily done by using the “margins” Stata post-estimation command to get predicted costs. This would help us see if the costs were truly different by wave after controlling for other covariates.

- To estimate differences by wave for each characteristic (Figure 3A), authors may want to include both wave categorical variable and its interaction with those characteristics (eg wave (1,2,3), age (<12, 12-19, 20-29, 30-39, 40-49, 50-59, 60-69, 70-79, 80), and wave*age). They can then use the Stata “margins” post-estimation command to predict the cost in each wave and at each level of covariate included in the interaction.

 

Reviewer #2: 

I appreciate the work of authors but there are some issue which in my view do not match.

The major thing is that the title, the abstract and the conclusions suggest that achieved results characterize the cost/patient for entire population of COVID-19 patient in Spain. The sample selection and thus the methods and the results do not support this hypothesis. In my view there two options (1) if the authors try to extend the costs results for a typical ICU/non-ICU patient from Spain, then they need a representative sample; (2) if the authors cannot achieve a representative sample, I recommand to underline the fact, that this is work is just a case-study, and everywhere in the paper should be clearly specified that "in our/current sample"/ "in our/current case", the results are. The phrases which induce the extension of results should be avoided in my view. Indeed in the limitations paragraph the authors try to explain differences between the current sample and the population, but the arguments are not solid. My worries are related to the thing that costs may be overestimated in particular hospitals compared with anothers. There is no proof that error are compensating each-other.

Response to reviewer #2:

Many thanks to the reviewer for his comments as we consider that they have improved the manuscript. The changes implemented in response to the concerns raised by the reviewer are included below.

We have introduced the following changes in different sections of the manuscript: 

- Title of the manuscript to: “Healthcare resource use and associated costs in a cohort of hospitalized COVID-19 patients in Spain: a retrospective analysis from the first to the third pandemic wave. EPICOV study.”

- Conclusions section from the Abstract (line 39): “LOS was longer for patients admitted to the ICU (especially in the first two waves) and for older patients in our study cohort; these populations incurred the highest hospitalization costs.”

- Introduction section (lines 81-82): “In the present study, we aimed to use real-world data to describe the use of healthcare resources and the associated costs in a cohort of hospitalized patients due to SARS-CoV-2 during three different outbreak waves from the perspective of Spain's national health system (NHS).”

- Discussion section:

Lines 396-396: “We have described the use of healthcare resources and costs in a cohort of hospitalized COVID-19 patients in Spain”

Line 432: “We should also point out that mean costs per patient and per day in our study cohort did not differ substantially between the outbreak waves”

Lines 449-450: “In addition, the treatments administered to patients differed drastically between the first wave and the other two. In this respect, during the first wave, most patients in our sample were treated with aminoquinolines and antivirals”

Lines 458-459: “Another interesting observation was that mean costs per COVID-19 patient and, especially, the LOS increased with age in our study cohort.”

Lines 464: “The results of the regression model confirm that length of stay is one of the main factors influencing cost in both ICU and non-ICU populations of our study sample.”

Lines 486-492: “Our cost analysis has some limitations that should be mentioned. First, the data were extracted from the HM Hospitals database, which agglutinated data from a consortium of private hospitals in Spain. Furthermore, this sample of hospitals represented 2.2% of Spanish hospitals. In that respect, the study population might not accurately represent the population attended in the Spanish healthcare system. Nonetheless, we consider that our results could be reasonably representative of the Spanish healthcare system at the population and cost-analysis level…”

A reference has been included to support the included text: Hospital statistics - National tables (2020). [Cited 7 December 2022]. In: Spanish Ministry of Health [Internet]. Available from: https://www.sanidad.gob.es/estadEstudios/estadisticas/docs/TablasSIAE2020/Tablas_Nacionales_2020.pdf

Regarding this point, we would like to point out that a sensitivity analysis was carried out taking into account a possible under- or overestimation of the cost.

Lines 506-508: “Therefore, we suggest that the costs obtained are an approximation of the actual costs of the Spanish public healthcare system.”

- Conclusions section:

Lines 530-534: “This study describes the use of healthcare resources and costs in a cohort of hospitalized COVID-19 patients over the course of the first year of the pandemic. Accordingly, our main observation is that mean costs per hospitalization COVID-19 patient did not consistently differ across the three main outbreak waves in our sample, especially for those patients who were not admitted to the ICU”

Lines 540-541: “In summary, our cost analysis provides robust data that could be useful to inform decision-makers about how health systems were impacted across the different outbreak waves in the first year of the COVID-19 pandemic.”

---

## [Decision Letter · Decision Letter 2]

12 Jan 2023

Healthcare resource use and associated costs in a cohort of hospitalized COVID-19 patients in Spain: a retrospective analysis from the first to the third pandemic wave. EPICOV study.

PONE-D-22-09612R2

Dear Dr. Drago,

We’re pleased to inform you that your manuscript has been judged scientifically suitable for publication and will be formally accepted for publication once it meets all outstanding technical requirements.

Kind regards,

Martial L Ndeffo Mbah, Ph.D

Academic Editor

PLOS ONE

Additional Editor Comments (optional):

Reviewers' comments:

Reviewer's Responses to Questions

**Comments to the Author**

1. If the authors have adequately addressed your comments raised in a previous round of review and you feel that this manuscript is now acceptable for publication, you may indicate that here to bypass the “Comments to the Author” section, enter your conflict of interest statement in the “Confidential to Editor” section, and submit your "Accept" recommendation.

Reviewer #2: (No Response)

2. Is the manuscript technically sound, and do the data support the conclusions?

Reviewer #2: No

3. Has the statistical analysis been performed appropriately and rigorously? 

Reviewer #2: No

4. Have the authors made all data underlying the findings in their manuscript fully available?

Reviewer #2: Yes

5. Is the manuscript presented in an intelligible fashion and written in standard English?

Reviewer #2: (No Response)

6. Review Comments to the Author

Reviewer #2: Furthermore, this sample of hospitals represented 2.2% of

489 Spanish hospitals [41]. In that respect, the study population might not accurately represent

490 the population attended in the Spanish healthcare system. Nonetheless

Therefore, we suggest that

507 the costs obtained are an approximation of the actual costscosts were considered

508 representative of the Spanish public healthcare system.

If you assume that your patients sample might not accurately represent the entire population of patients, then in my view there are not enough suport information to conclude that estimated cost are a bad or a good representation of the real costs. I really appreciate your efforts but I cannot accept the conclusion without a strong evidence report.

7. PLOS authors have the option to publish the peer review history of their article (what does this mean?). If published, this will include your full peer review and any attached files.

Reviewer #2: No

---

## [Editor Report · Acceptance letter]

16 Jan 2023

PONE-D-22-09612R2 

Healthcare resource use and associated costs in a cohort of hospitalized COVID-19 patients in Spain: a retrospective analysis from the first to the third pandemic wave. EPICOV study. 

Dear Dr. Drago:

I'm pleased to inform you that your manuscript has been deemed suitable for publication in PLOS ONE. Congratulations! Your manuscript is now with our production department. 

Kind regards, 

on behalf of

Dr. Martial L Ndeffo-Mbah 

Academic Editor

PLOS ONE